# Mechanisms of Physical Exercise Effects on Anxiety in Older Adults during the COVID-19 Lockdown: An Analysis of the Mediating Role of Psychological Resilience and the Moderating Role of Media Exposure

**DOI:** 10.3390/ijerph20043588

**Published:** 2023-02-17

**Authors:** Shuangshuang Xin, Xiujie Ma

**Affiliations:** 1School of Wushu, Chengdu Sport University, Chengdu 610041, China; 2Chinese Guoshu Academy, Chengdu Sport University, Chengdu 610041, China

**Keywords:** COVID-19, older adults, physical exercise, anxiety, psychological resilience, media exposure

## Abstract

The purpose of this study was to explore the effects of physical exercise on anxiety in older adults during the COVID-19 pandemic lockdown, as well as the mediating role of psychological resilience and the moderating role of media exposure. An online questionnaire was used to survey older adults in Chengdu, Southwest China. A total of 451 older adults aged 60 years and older participated in the study (209 males and 242 females). The results suggest that physical exercise negatively influenced anxiety symptoms in older adults; psychological resilience mediated the effect of physical exercise on anxiety in older adults and negatively predicted it; furthermore, media exposure moderated the effects of physical exercise and psychological resilience on anxiety, and low levels of media exposure strengthened these effects. This study suggests that participation in physical exercise and reduced media exposure during the COVID-19 pandemic lockdown may have reduced anxiety in older adults.

## 1. Introduction

The COVID-19 pandemic has become the largest public health crisis worldwide [1], with more than 600 million cumulative COVID-19 infections and more than 6.5 million deaths worldwide as of October 2022, and the numbers have continued to increase. Older adults represent a group that was highly infected by COVID-19 and that was more vulnerable during the pandemic [2]. Data from a published study showed that 3.6% of people aged 60 years and older and from 8% to 14.8% of people in their 70s and 80s were at risk of dying from COVID-19 [3]. The pandemic has led to fundamental lifestyle changes in older adults, such as a decrease in physical exercise and socialization [4], an increase in sedentary time [5], and an increase in psychological disorders [6], which have a negative influence on their physical and mental health. Studies have shown that insecurity, due to the death and fear caused by the COVID-19 pandemic, further exacerbates anxiety symptoms in older adults [7,8]. In addition, studies from several countries have noted that the COVID-19 pandemic significantly increased anxiety levels in older adults [9,10,11,12,13]. In particular, during the lockdown of the COVID-19 pandemic, people were required to practice home isolation, which exacerbated anxiety symptoms [14].

In September 2022, the COVID-19 pandemic spread on a large scale in Chengdu, China, and to prevent further spread, the government took a series of pandemic prevention and control measures, such as the lockdown of the entire city, quarantining at home, limiting people’s travel, and maintaining social distance. This largely affected the lifestyle of older adults and had a serious impact on their mental health. During this time, all people were required to self-isolate by staying at home, and home exercise became an important tool to promote mental health among older adults. Researchers pointed out that physical activities at home during isolation can help improve an individual’s mental health and alleviate symptoms such as anxiety and depression [5,15]. Therefore, this study was based on the context of the lockdown during the COVID-19 pandemic to reveal the effects of physical exercise on anxiety symptoms in older adults and to explore the intrinsic mechanisms of the effects to provide a theoretical basis for promoting mental health in older adults under a pandemic.

### 1.1. Physical Exercise and Anxiety

Anxiety is a typical negative emotional state that can result from psychological stress. During the COVID-19 pandemic, anxiety was an unpleasant experience that people had in response to unexpected situations, manifested as anxiety about infection and health [16]. It was reported that excessive anxiety levels during the pandemic led to restlessness, and therefore, people became sick more easily. Particularly during the lockdown of the COVID-19 pandemic, people were forced into home quarantine, which led to severe anxiety problems due to difficulties in obtaining daily necessities, fear of infection, inability to cure, and current concerns about a vaccine for this infectious disease [17]. A Chinese study showed that older adults who experienced difficulties in their daily lives during the COVID-19 pandemic were 3.72 times more at risk for anxiety symptoms than those who did not experience difficulties in their daily lives during the pandemic [18].

Physical exercise is considered to be a positive medium for reducing anxiety symptoms, especially for significantly alleviating stress, anxiety, and other poor mental health states due to public emergent crisis events [19]. A study reported anxiety levels in 38 participants before and after tai chi and yoga exercises at home and found a significant trend towards a decrease in anxiety levels [20]. In addition, maintaining physical activity during the COVID-19 pandemic was reported to protect against viral infections of the respiratory tract, prevent reduced physical function, and slow down anxiety symptoms [21]. Several meta-analyses have indicated that engaging in physical exercise had a facilitative effect on reducing anxiety in non-clinical populations and in patients with anxiety disorders, and that physical exercise had no side effects associated with reducing anxiety symptoms in patients compared to conventional medication [18,22,23]. During the COVID-19 pandemic, a positive association between physical exercise and mental health in older adults was reported [24], particularly a strong association with anxiety, as evidenced by lower levels of anxiety with higher levels of physical exercise [25]. It was concluded that maintaining physical exercise during the COVID-19 pandemic would help to mitigate the post-pandemic side effects on mental health [26].

Thus, Hypothesis 1 was proposed:

**Hypothesis 1** **(H1).***Physical exercise negatively predicts anxiety levels in older adults*.

### 1.2. Physical Exercise, Psychological Resilience, and Anxiety

Psychological resilience is a psychological strength that every individual possesses for effectively negotiating, adapting, and managing stress or trauma; psychological resilience reduces the adverse effects of negative events and allows individuals to successfully cope with adversity [27]. The results of the lockdown, social distancing, and lack of exercise were shown to result in significantly increased levels of mental health concerns such as loneliness, depression, anxiety, and suicidal ideation [28]. Studies have shown that psychological resilience during the COVID-19 outbreak may have reduced the negative effects due to the COVID-19 pandemic, demonstrated by alleviating people’s anxiety during the lockdown [29] and maintaining mental health [30]. When individuals face threats, psychological resilience can provide protective factors that help people to effectively cope with stress and mitigate the negative effects of stress, and therefore, can promote physical and mental health, social adjustment, and quality of life [31]. During the COVID-19 pandemic, social isolation may have caused individuals to be more vulnerable to mental illness, but psychological resilience could act as a buffer to help individuals to maintain their mental health [32].

Research has shown that psychological resilience in older adults has a positive effect on reducing anxiety and promoting the development of mental health, and physical activity is the basis for improving psychological resilience [33]. Most older adults can overcome the challenges of daily life as well as anxiety through recreational physical exercise and other productive physical activities that can increase psychological resilience [34]. The positive effects of physical exercise on psychological resilience have been demonstrated by the ability to induce positive physical and psychological improvements, to prevent the effects of stressful events, and to prevent or minimize several neurological disorders [35]. During blockades, people who exercised more frequently have shown higher levels of psychological resilience and positive emotions and lower levels of negative emotions such as anxiety and depression [36].

Thus, research Hypothesis 2 was proposed: 

**Hypothesis 2** **(H2).***Psychological resilience plays a mediating role between physical exercise and anxiety in older adults*.

### 1.3. Media Exposure and Anxiety

The decline in mental health due to the COVID-19 pandemic is a hot issue of global concern, and Gao et al. suggested that exposure to news and information about the pandemic may have exacerbated fear and anxiety levels in the general population [37]. Further research has found that the more people were exposed to media coverage of COVID-19, the higher the level of anxiety they exhibited [38]. During the pandemic, people were often exposed to media to obtain information about the pandemic, which affected their perceptions. Global data suggest that repeated exposure to media coverage of traumatic events may produce immediate and long-term mental health problems, especially for those already amid a hazardous event [39].

A Chinese study found that 82% of participants had regular access to social media during the COVID-19 pandemic, and since a large number of Internet users expressed their negative emotions on social media, this could have influenced the anxiety of other Internet users to some extent [37]. Another study also showed that media coverage of pandemic-related content increased negative emotions and that this negative news may have caused anxiety, fear, anger, sadness, etc. [40]. A research study in the United States concluded that, for each additional hour spent on social media, there was a 0.14 percentage point increase in mental distress for each additional traditional media source [41]. Specifically, the longer the exposure to COVID-19 messages, the more likely people were to remember them, which could trigger negative emotional responses such as fear and anxiety [42]. Studies have found that high levels of media exposure increased the risk of anxiety disorders by 0.3 times compared to low levels of media exposure, and excessive media exposure could affect the level of public mental health [43]. Previous research has confirmed that media exposure affects people’s anxiety levels during the pandemic and that anxiety is directly influenced by physical activity and mental toughness; thus, this study used media exposure as a moderating variable to examine whether high or low levels of media exposure have an effect on physical activity and psychological resilience on anxiety.

Thus, research Hypothesis 3 was proposed: 

**Hypothesis 3** **(H3).***Media exposure has a moderating effect during the effect of physical exercise on anxiety*.

Additionally, Hypothesis 4 was proposed: 

**Hypothesis 4** **(H4).***Media exposure has a moderating effect during the effect of psychological resilience on anxiety*.

### 1.4. Hypotheses and Conceptual Model

Based on the above literature review and hypotheses, in this paper, a research model was constructed. As shown in Figure 1, this research model schematically shows that physical exercise has a direct effect on anxiety, psychological resilience plays a mediating role between physical exercise and anxiety, and media exposure plays a moderating role in the relationship between physical exercise and psychological resilience on anxiety.

## 2. Methods

### 2.1. Participants and Procedure

To study the mechanisms of physical exercise effects during the COVID-19 pandemic, in this study, we used a Chinese online questionnaire platform (WJX). Questionnaires were distributed to participants between September and October 2022. Participants were recruited through random sampling in the city of Chengdu in Southwest China. A total of 500 randomized questionnaires were distributed, and 451 valid questionnaires were returned, with a return rate of 90.20%. The demographic characteristics of the questionnaire were: age, gender, education, marital status, income level, and exercise program; see Table 1 for detailed data. The participants were all over 60 years of age (209 males and 242 females) with a mean age of 64 ± 1.25 years and were able to read, understand, and fill out the questionnaire. Inclusion criteria included: (1) the absence of major physical and mental illness, (2) informed consent, and (3) the absence of cognitive impairment. Exclusion criteria included: (1) repeated invalid questionnaires and (2) questionnaires that took less than 1 min to fill out. The study was approved by the ethical review committee of the institute, and participants were required to read the informed consent form before filling in the questionnaires and to click “agree” before starting to fill in the questionnaires. All questionnaires were filled out voluntarily by the participants, participants were allowed to withdraw during the process of filling out the questionnaires, and rewards were given to participants who filled out the questionnaires after the completion of the study. The data collected in this study are strictly confidential and cannot be released without the consent of the participants.

### 2.2. Instruments

#### 2.2.1. Physical Exercise

The physical exercise survey included individuals who participated in physical fitness activities during the COVID-19 pandemic in the last month (from 1 September 2022 to 1 October 2022) and the specific circumstances of participation in exercise, such as the items of participation in fitness exercise, exercise time, exercise frequency, and exercise intensity. The exercise level of older adults was scored comprehensively by using the exercise time, exercise frequency, and exercise intensity, and the scoring method referred to Liang’s “physical exercise level scale”. The score range of exercise time was 0–4, the score range of both exercise frequency and exercise intensity was 1–5, and the level of physical exercise = time × frequency × intensity [44]. The lowest score was 0, the highest score was 100, and the higher the score the higher the level of physical exercise. In this study, the Cronbach’s alpha coefficient was 0.861.

#### 2.2.2. Psychological Resilience

Psychological resilience was measured using the Conner–Davidson resilience scale (CD-RISC), which showed good psychometric properties, and the factor analysis yielded five factors, i.e., competence, tolerance of negative affect, acceptance of change, control, and spiritual influences [45]. This scale has been validated in several studies on Chinese older adults [46,47]. The scale consists of 25 items, each rated on a Likert 5-point scale ranging from 0 (not at all correct) to 4 (completely correct). Total scores ranged from 0 to 100, with higher scores indicating greater resilience. In the present study, the Cronbach’s alpha coefficient was 0.905.

#### 2.2.3. Anxiety

Anxiety was assessed using the generalized anxiety disorder 7-item (GAD-7) scale developed by Spitzer et al. [48], which has been widely used in the study of anxiety symptoms in Chinese older adults and has good validity [49,50]. The scale consists of seven symptoms, including “feeling restless, worried, and annoyed”, “unable to stop or control worry”, “worrying too much about various things”, “very tense and difficult to relax”, “very anxious so I can’t sit still”, “easily upset or easily angry”, and “feel as if something terrible has happened something terrible has happened”. The scores for these options are: 0 = never, 1 = a few days, 2 = more than half a day, and 3 = almost every day. The total score ranged from 0 to 21, with higher scores indicating higher anxiety symptoms. In the present study, the Cronbach’s alpha coefficient was 0.896.

#### 2.2.4. Media Exposure

Media exposure was measured by participants’ responses to concerns about COVID-19 information. Considering the media exposure habits of older adults in their daily life, the questionnaire asked the participants about their attention to new COVID-19 information on the radio and television, newspapers, WeChat, Shake, and short video sites. The options ranged from “very concerned” to “not concerned at all”, with a score from 5 to 1. The higher the score, the higher the level of exposure to COVID-19 information. In the present study, the Cronbach’s alpha coefficient was 0.836.

### 2.3. Control Variables

Since anxiety is a psychological state of individuals and is influenced by both personal and social factors, in this study, gender, age, education, marriage, and income were used as control variables to reduce the influence of irrelevant variables on the study results.

### 2.4. Analysis

Data from this study were organized and analyzed using the SPSS 25.0. AMOS 24.0 and the SPSS macro program PROCESS 3.4 developed by Hayes. All statistical tests were two-sided, i.e., a *p*-value < 0.05 was considered to be statistically significant. First, the common methodological bias problem was tested by Harman’s one-way test, and the results showed that the unrotated exploratory factor analysis yielded 26.89% (<40%) of the variance explained by the first one-way factor; therefore, there was no serious methodological bias problem in this study. Next, confirmatory factor analysis and model fit were performed by AMOS to demonstrate the validity of the individual variables of the model. Then, descriptive statistics of the main study variables were conducted, and a Pearson correlation analysis was used to determine the relationships among the variables. Then, we performed mediation and moderating effect tests using the bootstrap method. Finally, slope plots were drawn to further determine the effects of the moderating variables.

## 3. Results

### 3.1. The Test of Reliability and Validity

Exploratory factor analysis and confirmatory factor analysis were considered as a way to test the validity of the model variables [51]. As shown in Table 2, the compositional reliability among the variables ranged from 0.752 to 0.918, and the mean extracted variance ranged from 0.577 to 0.728. According to the studies of Chin (1998) [52] and Fornell (1985) [53] on CR and AVE, the values taken in this study were better, indicating that the variables had good reliability and validity. In addition, we also tested the fit of the model (Table 3), where GFI, AGFI, CFI, NFI, and IFI were all greater than 0.9 and RMSEA = 0.033, indicating a good fit of the model.

### 3.2. Descriptive Statistics and Correlations among the Main Study Variables

Descriptive and correlation analyses were performed on the main variables; Table 4 shows the means and SDs of the variables and the Pearson correlations among the variables. The results showed that physical exercise and anxiety were significantly negatively correlated and significantly positively correlated with psychological resilience; psychological resilience was significantly negatively correlated with anxiety; meanwhile, media exposure was significantly correlated with other variables.

### 3.3. The Mediation Model Analysis

To further validate the mediating effect of psychological resilience, in this study, we conducted a mediating effect test by estimating the mediating effect bootstrap 95% CI of 2000 samples. Table 5 shows that physical exercise negatively predicted anxiety (β = −0.351, *p* < 0.001), and therefore, Hypothesis 1 was supported; physical exercise positively predicted psychological resilience (β = 0.096, *p* < 0.001) and psychological resilience negatively predicted anxiety (β = −0.425, *p* < 0.001). Moreover, Table 6 displays that the direct effect of physical exercise on anxiety was −0.351, 95% CI = [−0.162, −0.095], and the indirect effect of psychological resilience was −0.166, 95% CI = [−0.087, −0.029]. The Bootstrap 95% CI for both direct and indirect effects did not contain 0, indicating that both effects reached a significant level. Therefore, the connection between physical exercise and anxiety was partially mediated by psychological resilience, and therefore, Hypothesis 2 was supported.

### 3.4. The Moderating Model Analysis

As shown in Table 7, physical exercise significantly negatively predicted anxiety (β = −0.351, t = −6.376, *p* < 0.001), further supporting Hypothesis 1, and physical exercise positively predicted psychological resilience (β = 0.092, t = 15.662, *p* < 0.001), psychological resilience negatively predicted anxiety (β = −0.434, t = −9.419, *p* < 0.001), and psychological resilience partially mediated the effect of physical exercise on anxiety, further supporting Hypothesis 2. In addition, the interaction term of physical exercise and media exposure predicted anxiety significantly (β = −0.021, t = 3.682, *p* < 0.001), suggesting that media exposure moderated the relationship between physical exercise and anxiety, and therefore, Hypothesis 3 was supported; meanwhile, the interaction term of psychological resilience and media exposure had a significant predictive effect on anxiety (β = −0.016, t = −2.368, *p* < 0.001), indicating that media exposure also moderated the relationship between psychological resilience and anxiety, and Hypothesis 4 was supported.

Finally, the nature of the moderating effect was further elucidated by simple slope analysis. Figure 2 shows that the negative predictive effect of physical exercise on anxiety in older adults was significant for those with low media exposure (1 SD below the mean, β simple = −0.22, *p* < 0.001) but was non-significant for those with high media exposure (1 SD above the mean, β simple = −0.02, *p* > 0.05). The effect of physical exercise on anxiety was more significant in older adults with low levels of media exposure relative to those with high levels of media exposure. That is, the effect of physical exercise on anxiety in older adults is enhanced as the level of media exposure decreases.

Figure 3 shows that the negative predictive effect of psychological resilience on anxiety in older adults was significant in high-level media exposure (1 SD above the mean, β simple = −0.08, *p* < 0.001); the negative predictive effect of psychological resilience on anxiety in older adults was enhanced in low-level media exposure (1 SD below the mean, β simple = −0.13, *p* < 0.001). The effect of psychological resilience on anxiety was more pronounced in older adults with low-level media exposure relative to those with high-level media exposure. That is, the effect of psychological resilience on older adults’ anxiety increases as the level of media exposure decreases.

## 4. Discussion

### 4.1. The Direct Effect of Physical Exercise

The results of this study showed that physical exercise had a significant negative predictive effect on anxiety symptoms in older adults (β = −0.351, *p* < 0.001) and that maintaining physical exercise during the COVID-19 pandemic could alleviate anxiety symptoms and could promote physical and mental health in older adults, which was consistent with previous studies [19,26,54]. Although outdoor physical exercise was not possible during the COVID-19 pandemic, given the positive effects of exercise on strengthening the immune system and relieving anxiety and depression, indoor exercise could still achieve a mental health-promoting effect [26]. In the present study, a large number of older adults reported participating in physical exercise with tai chi and yoga during the home isolation period of the pandemic. One study reported that resistance training, yoga, and tai chi could be used as alternative exercise therapies for anxiety, which had important implications for older adults who were unable to participate in strenuous or high-impact exercise such as jogging and other outdoor sports [55]. Since the population of older adults was highly infected with COVID-19, the fear of being infected was the main cause of their anxiety, and physical exercise is the best means to alleviate this concern. First, participation in physical exercise can divert the attention of older adults and improve cognitive function [56]; second, physical exercise has a positive effect on the cardiovascular and respiratory systems [57], which can better improve the resistance to COVID-19. Therefore, physical exercise during the pandemic lockdown has a positive contribution by alleviating anxiety symptoms in older adults.

### 4.2. The Mediation of Psychological Resilience

The results of the present study showed that psychological resilience partially mediated the relationship between physical exercise and older adults’ anxiety, with physical exercise significantly and positively predicting psychological resilience (β = 0.092, *p* < 0.001) and psychological resilience significantly and negatively predicting older adults’ anxiety (β = −0.434, *p* < 0.001). The results of this study were consistent with previous studies that showed that physical exercise increased the psychological resilience of older adults [33,34] and that the more psychologically resistant older adults could maintain a favorable attitude during the pandemic, face difficulties and challenges with ease, and could reduce the risk of developing anxiety disorders [29,30,58]. Studies have shown that as the level of physical exercise increased, psychological resilience also improved [59]. For older adults, regular participation in physical exercise is one of the most important indicators of improved psychological resilience [60]. First, long-term physical exercise improves the nervous system, increases the brain’s cognitive reserve for stress, increases resilience against risk, and in the face of crisis events (e.g., pandemic), the brain gives physical and psychological protection based on previous cognitive reserves, thus reducing the risk of chronic diseases as well as psychological disorders [35]. Second, in terms of emotional feelings in older adults, Rutten argued that positive emotional experiences were important components of psychological resilience [61]. Therefore, participation in physical exercise can generate positive emotional experiences, and positive emotions can positively affect the nervous system and can increase the cognitive, emotional, and psychological resilience of individuals to cope with stressful events in their lives [62,63].

Older adults with high levels of psychological resilience can reduce negative factors (e.g., anxiety, depression, etc.) in the objective environment by changing their environment or making selective changes to their environment and other proactive coping methods (e.g., physical exercise), thus protecting mental health states. Further research has found that psychological resilience was influenced by personal experience, and because older adults experience more social risks, especially major public health risk events, they may have higher psychological stress tolerance [64] and may be able to effectively moderate their psychological state in the face of a pandemic. Therefore, psychological resilience can be used as a moderating or buffering factor to provide “bounce-back” power in the face of adversity, and older adults with greater psychological resilience are better able to maintain physical and mental health and, therefore, reduce anxiety caused by stressful events. Therefore, this study concludes that participation in physical exercise by older adults can increase their confidence in fighting the pandemic, can accumulate psychological resources for them, and can improve their psychological resilience, which not only increases their resistance, but also reduces mental illness and the risk of anxiety disorders.

### 4.3. The Moderating of Media Exposure

The present study found that the effects of both physical exercise and psychological resilience on older adults’ anxiety symptoms were moderated by media exposure, and the effects of physical exercise and psychological resilience on older adults’ anxiety symptoms were stronger at low levels of media exposure as compared with those at high levels of media exposure, indicating that low levels of media exposure were more significant in reducing older adults’ anxiety symptoms, which is consistent with previous studies [37,38,41].

The present study showed that media exposure moderated the direct path of the model (β = −0.021, *p* < 0.001). Specifically, the negative effect of physical exercise on older adults’ anxiety symptoms was more pronounced at low levels of media exposure, but high levels of media exposure buffered this effect. First, older adults with high-level media exposure spent a lot of time on the media, which led to a decrease in their physical exercise time and increased sedentary time [5]. In addition, media-dependency theory suggests that audiences use media to obtain specific gratification or achieve certain goals, and if they lack other alternatives or resources, they become dependent on mass media [65]. Specifically, homebound older adults have reduced social activities, and access to information about the pandemic is mainly substituted through the media, thus significantly increasing their media consumption time. Secondly, media reports during the pandemic were mostly negative news about the pandemic, and prolonged exposure to such negative information may lead to negative emotions and lower self-efficacy [66]. Self-efficacy is an important factor influencing the participation of older adults in physical exercise; decreased self-efficacy has been shown to significantly reduce their level of physical activity [67]. Therefore, high levels of media exposure may reduce older adults’ exercise levels, thus attenuating the negative effect of physical exercise on older adults’ anxiety symptoms, while low levels of media exposure significantly enhance the negative effect of physical exercise on older adults’ anxiety symptoms.

In addition, this study found that media exposure also moderated the relationship between psychological resilience and anxiety in older adults (β = −0.016, *p* < 0.001). Psychological resilience is a personality trait of individuals and is considered to be a resource of stability that can be well-regulated when resisting external disturbances and stress [68]. Therefore, in an environment with high-level media exposure, older adults can effectively use their experience and knowledge to regulate their emotions. Since the pandemic has been in people’s lives for two years, people have gradually adapted to life under the pandemic and have actively taken protective measures, especially under the strong pandemic prevention and control measures in China, and the government’s organized, planned, and continuous dynamic management process has given older adults enough confidence to resist the stress caused by the pandemic [69]. Thus, psychological resilience can play a stronger role in helping older adults to maintain healthy emotions in a high-level media exposure environment, and its impact on older adults’ anxiety symptoms is further enhanced in a low-level media exposure environment.

In conclusion, the anxiety levels of older adults are lower in all low levels of media exposure. According to the agenda-setting theory, the media influences what people pay attention to and how they think and make decisions through agenda setting [70]. Therefore, in the context of the COVID-19 pandemic, the media influences people’s risk perception of the severity of the pandemic through selective presentation, opinion output, and value guidance, which, in turn, influences people’s anxiety symptoms. Excessive media exposure may lead to psychological problems in older adults because older adults in a pandemic may develop a negative psychological state out of concern for their health, and high levels of media exposure, where information about the pandemic is disorganized and rumors increase, may lead to severe anxiety in older adults. As scholars have stated, publishing the most up-to-date and accurate information, such as the number of people whose conditions have improved and the progress of medications and vaccines, can reduce anxiety levels [71]. In addition, mental health professionals recommend increasing healthy exercise behaviors and avoiding exposure to negative news [72].

### 4.4. Limits and Implications

This study has several limitations. First, this study is a cross-sectional study that does not effectively explain the causal relationships among the variables. Second, the physical exercise scale and media exposure scale in this paper were adapted and designed based on previous questionnaires, which have some limitations in applicability and need to be validated by more studies in the future. Third, when testing physical exercise, it was not measured by sport, but all sports were collectively referred to as physical exercise; future studies could be conducted for a particular exercise. Fourth, the sample size of this study is small, which greatly limits the representativeness of the article, and future studies could add more samples. Finally, the current study only considered the mediating role of psychological resilience and the moderating role of media exposure, and it is unclear whether there are other influences between physical exercise and anxiety in older adults that should be further explored in future studies, such as social support.

Despite these limitations, the present study yields theoretical and practical implications. On the one hand, this study explored the negative predictive effect of physical exercise on older adults’ anxiety during the COVID-19 outbreak and proposed the mediating role of psychological resilience in reducing older adults’ anxiety, providing theoretical guidance for enhancing older adults’ mental health. On the other hand, the effects of media exposure during the pandemic in moderating the effects of physical exercise and psychological resilience on anxiety in older adults clarified that reducing media exposure during the pandemic has practical implications for reducing anxiety symptoms in older adults. Therefore, this study encourages older adults to be physically active, not only as an important means to promote physical and mental health, but also as an important process to achieve healthy aging.

## 5. Conclusions

This study investigated a model that mediated the relationship between physical exercise and anxiety in older adults during the COVID-19 pandemic lockdown. The results indicated that physical exercise negatively influenced older adults’ anxiety, that psychological resilience mediated the relationship between physical exercise and older adults’ anxiety, and that media exposure moderated the effects of physical exercise and psychological resilience on older adults’ anxiety. Older adults with low levels of media exposure reported lower anxiety compared to those with high levels of media exposure. Thus, the effect of media exposure on anxiety in older adults is critical. Older adults should increase their physical exercise during the COVID-19 pandemic lockdown and should reduce paying excessive attention to pandemic-related information, especially negative information; we recommend that older adults pay more attention to positive events during a pandemic. Meanwhile, news media should provide information related to COVID-19 objectively and scientifically. In addition, older adults should deepen their knowledge of pandemic prevention and control and should enhance their perceptions of the controllability of the pandemic. Finally, the government should increase control of relevant media and regularly release information about the novel coronavirus, publicize the important role of physical exercise during the pandemic, promote effective home physical exercise methods, and organize various forms of exercise programs to guide older adults in the science of fitness to improve their organism resistance and, ultimately, to overcome the COVID-19 pandemic.

## Figures and Tables

**Figure 1 ijerph-20-03588-f001:**
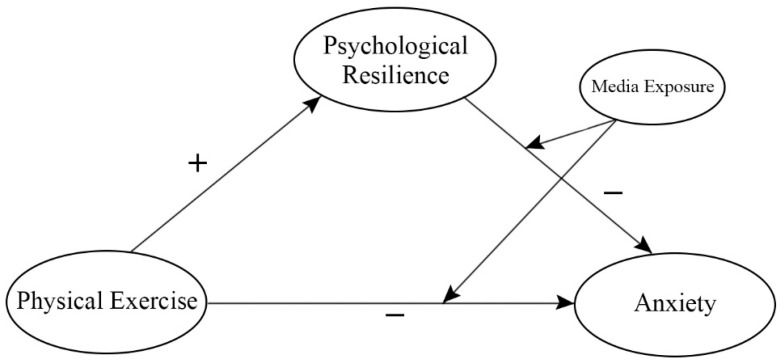
Conceptual framework.

**Figure 2 ijerph-20-03588-f002:**
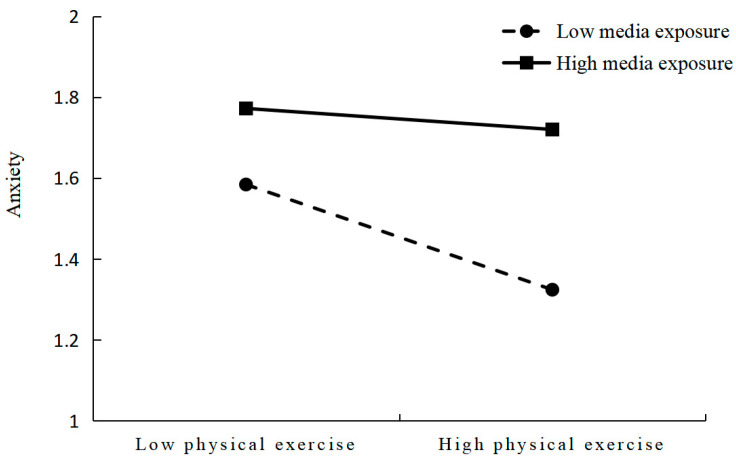
Interaction between physical exercise and media exposure.

**Figure 3 ijerph-20-03588-f003:**
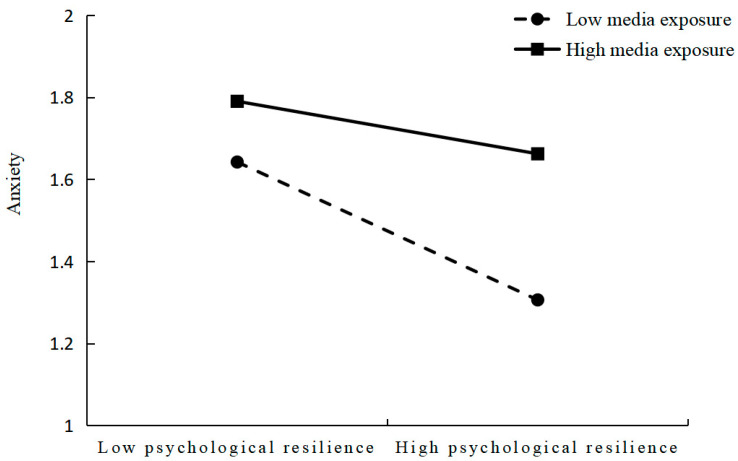
Interaction between psychological resilience and media exposure.

**Table 1 ijerph-20-03588-t001:** Demographic characteristics of the samples.

Variable	Frequency	Percentage (%)
Gender		
male	209	46.34%
female	242	53.66%
Age		
60–65	193	42.79%
65–70	127	28.16%
70–75	79	17.52%
75<	52	11.53%
Income		
<2000	119	26.39%
2000–3000	142	31.49%
3000–4000	71	15.74%
4000–5000	51	11.31%
>5000	68	15.08%
Education		
primary school	101	22.39%
high school	261	57.87%
college and higher	89	19.73%
Marital status		
married	403	89.36%
unmarried	1	0.22%
divorced	26	5.76%
widowed	21	4.66%
Program		
tai chi	127	28.16%
yoga	59	13.08%
qigong	106	23.50%
aerobics	118	26.16%
others	41	9.09%

**Table 2 ijerph-20-03588-t002:** Confirmatory factor analysis.

Variable	CR	AVE
Physical exercise	0.752	0.577
Psychological resilience	0.918	0.728
Media exposure	0.867	0.673
Anxiety	0.915	0.719

Note: CR = composite reliability; AVE = average variance extraction.

**Table 3 ijerph-20-03588-t003:** Model fit indices.

	Χ^2^/df	P	GFI	AGFI	CFI	NFI	IFI	RMSEA
Indices	1.629	0	0.948	0.938	0.953	0.942	0.953	0.033

**Table 4 ijerph-20-03588-t004:** Correlations and descriptive statistics of the main variables.

Variable	M	SD	1	2	3	4
1. Physical exercise	2.657	0.865	1			
2. Psychological resilience	3.201	0.980	0.269 ***	1		
3. Media exposure	2.569	0.842	−0.194 **	−0.481 **	1	
4. Anxiety	3.935	0.924	−0.301 ***	−0.237 ***	0.425 ***	1

Note: ** *p* < 0.01 and *** *p* < 0.001.

**Table 5 ijerph-20-03588-t005:** Regression analysis of the mediation model.

Predictors	Step 1 (Psychological Resilience)			Step 2 (Anxiety)		
	β	SE	t	β	SE	t
Physical exercise	0.096	0.024	2.395 ***	−0.351	0.042	2.977 ***
Psychological resilience				−0.425	0.063	−11.265 ***
*R* ^2^	0.589			0.503		
*F*	59.642 ***			72.163 ***		

Note: *** *p* < 0.001.

**Table 6 ijerph-20-03588-t006:** Bootstrapping analysis of the mediation model.

	Effect	SE	95% CI	Ratio to Total Effect
Direct effect	−0.351	0.042	[−0.162, −0.095]	69.39%
Indirect effect	−0.040	0.035	[−0.087, −0.029]	30.61%
Total effect	−0.391	0.057	[−0.155, −0.073]	-

**Table 7 ijerph-20-03588-t007:** Regression analysis of the moderating model.

Variable	M: Psychological Resilience			Y: Anxiety		
	β	SE	t	β	SE	t
Constant	−0.768	0.263	−3.967 **	−0.668	0.231	−4.632 **
Gender	−0.263	0.096	−3.755 *	−0.212	0.103	−0.339 **
Age	0.023	0.042	2.683	0.036	0.038	0.569
Physical exercise	0.092	0.026	12.662 ***	−0.351	0.105	−6.376 ***
Psychological r esilience				−0.434	0.069	−8.419 ***
Media exposure				0.634	0.065	10.684 ***
Physical exercise × media exposure				0.021	0.033	3.682 **
Psychological resilience × media exposure				0.016	0.024	−2.368 ***
*R* ^2^	0.526			0.497		
*F*	65.653 ***			70.391 ***		

Note: * *p* < 0.05, ** *p* < 0.01, and *** *p* < 0.001.

## Data Availability

The data presented in this study are available upon request from the corresponding author. The data are not publicly available due to an ethical agreement with the Chengdu Sport University Social Sciences Ethics Panel, to keep them under Ma Xiujie’s personal OneDrive account, which is not accessible to the public.

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
