# Peer review of "Mechanisms of Physical Exercise Effects on Anxiety in Older Adults during the COVID-19 Lockdown: An Analysis of the Mediating Role of Psychological Resilience and the Moderating Role of Media Exposure"

_ijerph, 2023, doi:10.3390/ijerph20043588_

Round 1
Reviewer 1 Report
It is an interesting, well-written and coherent work. The negative effects of the COVID-19 pandemic on the mental health of the elderly has been extensively researched and this paper contributes to understanding how physical exercise would be related to less anxiety. I suggest minor changes.
- Delete Figure 1 (The research model) since it is usually illustrated with a figure when the proposed analysis is a structural equation model (SEM). It is very clear and well justified the hypotheses proposed.
- Add in the limitations of the study the fact that the sample is small and not representative.
Author Response
Thank you for your kind comments on our article, which were very precise and helpful for us. In accordance with your suggestions, we have made a substantial revision of this article. Below, you will find a breakdown of the responses to your comments (in italics):
- Delete Figure 1 (The research model) since it is usually illustrated with a figure when the proposed analysis is a structural equation model (SEM). It is very clear and well justified the hypotheses proposed.
- Response: Thank you very much for your suggestion. We have improved this part
- Add in the limitations of the study the fact that the sample is small and not representative.
-Response: Thank you very much for your valuable suggestion, we have added to the restrictions section: Fourth, the sample size of this study is small, which greatly limits the representativeness of the article, and future studies could be added more samples.
We hope you are pleased with the revised manuscript, which also includes the changes expected from other peer reviewers.
Thanks again, and kind regards.
.
Reviewer 2 Report
My comments are provided below:
- The English itself is not bad or hard to comprehend. However, the style must be improved specially in the introduction. Sequence of ideas/sentences within a paragraph are not always, some of the sentences are very wordy while others are a little more difficult to understand. Examples: (1) the first paragraph does not have a clear flow of ideas... it is ok, but it seems to contain too many pieces of information in the same paragraph. (2) Strange sentence (lines 51 - 54): Researchers have pointed out that any physical exercise during home isolation could help to improve the mental health of individuals and could have a positive effect on the alleviation of mental illnesses such as anxiety and depression [6, 16]. (3) Too wordy (lines 54 - 57): Physical exercise played an important role during the lockdown period, and studies have shown a significant increase in the frequency of home exercise during the pandemic lockdown [17], which provided a relatively stable research setting and a large sample size for this study.
- Hypotheses 3 & 4 did not follow the same pattern as the first two. While there were long explanations for the first hypotheses, nothing is said about why you decided on H3 and H4.
- The model at the beginning of the methods should have been included in the introduction.
- A better explanation of how random sampling was used to recruit participants is necessary. Did the authors have access to the names of all older residents in Chengdu? Are all 60+ residents in Chengdu internet users? If it was random sampling, how many denied to participate in the study?
- Table 1: formatting needs a little bit of improvement (better separation of the variables)
- The biggest problem with this research was the use of instruments that did not seem to have been validated. Besides for internal consistency, it is not mentioned the title of the Physical Exercise questionnaire and validation evidence. The authors also do not describe validity evident for using the anxiety questionnaire with older Chinese adults. There was also no title or validity evidence presented to the Media Exposure questionnaire. If the instruments are not collecting information appropriately, then it is difficult to know how accurate the results of the study are.
The discussion was generally well written and is adequate considering the results of the study.
Author Response
Thank you for your kind comments on our article, which were very precise and helpful for us. In accordance with your suggestions, we have made a substantial revision of this article. Below, you will find a breakdown of the responses to your comments (in italics):
- The English itself is not bad or hard to comprehend. However, the style must be improved specially in the introduction. Sequence of ideas/sentences within a paragraph are not always, some of the sentences are very wordy while others are a little more difficult to understand. Examples: (1) the first paragraph does not have a clear flow of ideas... it is ok, but it seems to contain too many pieces of information in the same paragraph. (2) Strange sentence (lines 51 - 54): Researchers have pointed out that any physical exercise during home isolation could help to improve the mental health of individuals and could have a positive effect on the alleviation of mental illnesses such as anxiety and depression [6, 16]. (3) Too wordy (lines 54 - 57): Physical exercise played an important role during the lockdown period, and studies have shown a significant increase in the frequency of home exercise during the pandemic lockdown [17], which provided a relatively stable research setting and a large sample size for this study.
- Response: Thank you very much for your suggestions. (1) We have simplified the first paragraph. (2) We have rewritten the sentence: Researchers have pointed out that physical activities at home during isolation can help improve an individual’s mental health and alleviate symptoms such as anxiety and depression. (3) We have simply processed this sentence.
- Hypotheses 3 & 4 did not follow the same pattern as the first two. While there were long explanations for the first hypotheses, nothing is said about why you decided on H3 and H4.
- Response: Thank you very much for your suggestion. We explained it in the article: Previous researches have confirmed that media exposure affects people's anxiety levels during the pandemic, and that anxiety is directly influenced by physical activity and mental toughness, so this study used media exposure as a moderating variable to ex-amine whether high or low levels of media exposure have an effect on physical activity and psychological resilience on anxiety.
- The model at the beginning of the methods should have been included in the introduction.
- Response: Thank you very much for your suggestion. We have included the model in the introduction section.
- A better explanation of how random sampling was used to recruit participants is necessary. Did the authors have access to the names of all older residents in Chengdu? Are all 60+ residents in Chengdu internet users? If it was random sampling, how many denied to participate in the study?
- Response: Thank you very much for your suggestion. We did not collect the names of the participants because of the protection of the privacy of the participants in this study and because the names of the participants were not part of the study; in addition, most elderly people are now using WeChat, and the survey procedure of Questionnaire Star is simple and can be completed by elderly people; finally, a total of 500 random questionnaires were set up in this study, and 451 valid questionnaires were returned, and the remaining 49 questionnaires were considered invalid.
- Table 1: formatting needs a little bit of improvement (better separation of the variables)
- Response: Thank you very much for your suggestion and we have made appropriate changes to the table format.
- The biggest problem with this research was the use of instruments that did not seem to have been validated. Besides for internal consistency, it is not mentioned the title of the Physical Exercise questionnaire and validation evidence. The authors also do not describe validity evident for using the anxiety questionnaire with older Chinese adults. There was also no title or validity evidence presented to the Media Exposure questionnaire. If the instruments are not collecting information appropriately, then it is difficult to know how accurate the results of the study are.
- Response: Thank you very much for your valuable suggestion. The name of the physical exercise questionnaire is "physical exercise level scale", which is designed by Liang (1994). We marked the name in the article. The generalized anxiety disorder 7-item (GAD-7) scale has been widely used in the study of anxiety symptoms of the elderly in China, and it has good validity. We added references to the article. The media exposure scale is our self-made scale, so we conducted exploratory factor analysis and confirmatory factor analysis to check the validity of the scale. In addition, we have verified the validity of each variable of the research model.
Liang, D. Stress level of college students and its relationship with physical exercise. Chinese Mental Health Journal 1994, 8, 5-6.
Round 2
Reviewer 2 Report
I would like to thank the Authors for improving the paper. It reads a lot better. My concern is still with the instruments used, but at least the paper reads well and the limitation for the validity of the instruments is listed.